cryptography

cryptocurrencies, blockchain, wallets

**Author for correspondence:**
J. Herrera-Joancomartí
e-mail: jordi.herrera@uab.cat

One contribution to the 'Blockchain Technology special collection'.

# Another coin bites the dust: an analysis of dust in UTXO-based cryptocurrencies

C. Pérez-Solà[1,2,3], S. Delgado-Segura[1,2], G. Navarro-Arribas[1,2] and J. Herrera-Joancomartí[1,2]

[1]Department of Information and Communications Engineering, Universitat Autònoma de Barcelona, Barcelona, Catalonia
[2]CYBERCAT, Centre de recerca en ciberseguretat de Catalunya, Tarragona, Catalonia
[3]Department of Computer Engineering and Mathematics, Universitat Rovira i Virgili

CP-S, 0000-0001-7534-1326; SD-S, 0000-0001-7783-7288;
GN-A, 0000-0003-3535-942X; JH-J, 0000-0002-4935-4681

Unspent Transaction Outputs (UTXOs) are the internal mechanism used in many cryptocurrencies to represent coins. Such representation has some clear benefits, but also entails some complexities that, if not properly handled, may leave the system in an inefficient state. Specifically, inefficiencies arise when wallets (the software responsible for transferring coins between parties) do not manage UTXOs properly when performing payments. In this paper, we study three cryptocurrencies: Bitcoin, Bitcoin Cash and Litecoin, by analysing the state of their UTXO sets, that is, the status of their sets of spendable coins. These three cryptocurrencies are the top-3 UTXO-based cryptocurrencies by market capitalization. Our analysis shows that the usage of each cryptocurrency presents some differences, and led to different results. Furthermore, it also points out that the management of the transactions has not always been performed efficiently and therefore, the current state of the UTXO sets is far from ideal.

## 1. Introduction

Blockchain-based cryptocurrencies are built on top of an append-only ledger shared among all the users of the system. Once a transaction is included in the ledger, the transaction can no longer be modified. Transactions aim to modify account balances.

How such accounts are stored and their balances modified is implemented in entirely different ways depending on the specific blockchain technology used. The two main approaches are Unspent Transaction Output (UTXO)-based and account-based, and the most relevant cryptocurrencies for each category are Bitcoin and Ethereum, respectively.

In the UTXO-based approach, currency units can be identified as coins represented by a specific data-structure known as UTXO.

As the coins of the system, the set of all UTXOs represents the total currency supply. Each coin contains essentially two different pieces of data: ownership data (specifically, the conditions that have to be fulfilled to spend it), and the amount it represents (that is, the units of the currency it symbolizes). Therefore, in this model coins do not have a fixed amount: a single coin may represent just the smallest unit of the currency (for instance, 1 satoshi in Bitcoin) up to the total currency supply (e.g. a unique coin with 21 million bitcoins).

On the other hand, in the account-based approach, the currency is spread among the multiple accounts of the system, each of which has a specific balance. The sum of all accounts' balances is the total amount of the cryptocurrency in circulation. With this approach, the system has to track the balance of every account, even if such balance is zero. Notice that in this case, the data required to operate the cryptocurrency is bound to the total number of accounts rather than the total number of coins.

Both models present their benefits and problems in terms of efficiency, scalability, decentralization and privacy.

One of the benefits of the UTXO model is its atomicity. Transactions can be processed in parallel for different UTXOs. There is no need to order transactions since coins involved in one transaction do not affect other coins of the system. An entity may own different coins and they can all be used alongside. The UTXO model also entails some benefits regarding privacy. Coins are attached to an address, that can be different for each coin. Although such approach may be also used in the account-based approach, this model suggests holding a single address per user, to minimize the number of zero-balance accounts the system has to keep track of. By contrast, a UTXO approach may discard the UTXO representing a coin once such coin has been spent. Furthermore, atomic transactions involving inputs from different parties are also easier to perform with a UTXO model than with an account-based model [1].

However, the UTXO approach comes with some shortcomings, its complexity being the main one. When a user needs to know the total value she owns, it is much easier to check her balance directly than to collect all the coins she owns and count them. Furthermore, for a simple payment transaction where user $A$ pays an amount $x$ to user $B$, the account-based model can perform such operation efficiently but the UTXO approach may generate some inefficiencies if $A$ does not have a coin with the exact amount $x$. For instance, if $A$ has a coin with an amount greater than $x$, the transaction will increase the number of coins in the system since the initial coin from $A$ will be transformed in two new coins, a coin of $x$-value for $B$ and another coin for the remainder value to $A$ (the change). On the other hand, if $A$ did not have a coin with amount greater or equal than $x$, she will have to collect multiple coins to make the payment. Such complexities that arise in a simple payment in the UTXO approach become more complex when other considerations enter into the equation. For instance, in order to reward miners, transactions include a fee and such fee is related to the size of the transaction. The key point here is that such fee is independent of the value the transaction is transferring, so surprisingly, transferring 1000 bitcoins may incur the same fees as transferring 0.00001 bitcoins. This fact has huge implications regarding the use of coins. Note that such idea indicates that when performing a payment you will pay a fee depending on the total number of coins you use in that payment. So if you have to collect multiple coins to achieve the $x$-value of your payment, you will have to pay higher fees than if you had a single coin. Notice, however, that unless you have the exact value, after the payment you will have a coin with a value containing the change. After multiple payments with all your bigger coins, you will end up with multiple small coins. In the following payment, you do not have any other option that collecting multiple coins to perform the payment, paying a higher fee for that transaction.

The complexity even goes further when the coins that are generated in a transaction end up with a value that is lower than the fee needed to spend them. In this case, the newly created coins will probably not be spent, since it is more expensive to spend the coin than not using the coin at all. In that situation, the system becomes inefficient since it has to track the status of that particular coin, that may never be spent due to an economic disincentive to do so.

To avoid such inefficiencies and to minimize the fees that users pay for their transactions, wallets should perform accurate management of the UTXOs that they use for their payments. However, as we will see in the analysis presented in this paper, such efficient management has not always been performed by all wallets.

The main contribution of this paper is an analysis of the UTXO sets of the three most capitalized UTXO-based cryptocurrencies: Bitcoin, Bitcoin Cash and Litecoin.[1] The paper provides insightful details about both the current and the evolution of the status of the UTXO sets of those coins. This analysis allows also to compare the usage of the three cryptocurrencies. Moreover, the paper further analyses one of the problems of UTXO-based cryptocurrencies: the existence of outputs not worth

---

[1]See https://web.archive.org/web/20180227064549/https://coinmarketcap.com/.

spending. We propose two metrics to evaluate whether an output is worth spending and analyse the UTXO sets of the three above-mentioned coins in terms of these metrics.

The rest of the paper is organized as follows. Section 2 provides an overview of the UTXO set concept. Then, §3 describes the contents of the current UTXO sets of Bitcoin, Bitcoin Cash and Litecoin. After that, §4 describes the metrics used to evaluate if outputs in the UTXO set are worth spending, that is, dust and unprofitability. Section 5 analyses the UTXO sets taking into account the previously defined metrics and considering both the current state and their evolution over time. Finally, §6 presents the related work and §7 the conclusions of the paper.

## 2. The UTXO set

In a UTXO-based cryptocurrency, the Unspent Transaction Output (UTXO) set is the subset of transaction outputs that have not been spent at a given point in time. Whenever a new transaction is created, funds are redeemed from existing UTXOs, and new ones are created. Basically, transactions consume UTXOs (in their inputs) and generate new ones (in their outputs). Therefore, transactions produce changes in the UTXO set. Regarding UTXOs, they can be identified by their *outpoint*, a two field data structure containing the transaction ID and output index that created them, and they store two data fields: the amount they are holding, and the locking script (`scriptPubKey`) that specifies the conditions under which they can be redeemed.

The main purpose of the UTXO set is speeding up the transaction validation process. When a new block is appended to the blockchain, full nodes update their vision of the UTXO set, removing the outputs that have been spent in the block and adding the newly generated ones. Being all unspent transactions outputs stored in the set, there is no need to scan throughout the blockchain to check for double-spends, but just check if the inputs of a transaction can be found in the set.

The format, storage requirements and implementation of the UTXO set may differ depending on the specific implementation. However, their content must be consistent between all the implementations, since all of them need to be able to verify every transaction. For our analysis, we have chosen the Bitcoin Core implementation. However, equivalent results can be obtained by any other implementation that follows the UTXO set approach. The main benefit of such a choice is that Bitcoin Core's implementation is also followed by wallets used in Bitcoin Cash and Litecoin so that the same analysis tool can be used for all three of them. In such implementation, the UTXO set is stored in the chainstate, a LevelDB database that provides persistent key-value storage. Apart from the UTXO set, the chainstate database stores two additional values: the block height at which the set is updated and an obfuscation key that is used to mask UTXO data [2,3]. Such an obfuscation key is used to obtain a different file signature of the UTXO set files for every different wallet, in order to avoid false-positives with antivirus software.

In order to perform the analysis included in the paper, we have created *STATUS* (STatistical Analysis Tool for Utxo Set) [4], an open source tool that provides an easy way to access, decode and analyse data from the Bitcoin's UTXO set.[2] STATUS works with a chainstate folder as input and can perform two different types of analysis: a UTXO-based analysis where each output is independent of the rest, and a transaction-based analysis where outputs belonging to the same transactions are aggregated. Moreover, comparative analysis between different states of the UTXO set can be also performed by using several chainstate snapshots. Our analysis has been complemented using BlockSci [5] to obtain additional data that cannot be covered by a static analysis of the UTXO set, such as unprofitability outputs estimation, as we will see later in §4.

## 3. Analysis of the UTXO sets of Bitcoin, Bitcoin Cash and Litecoin

In this section, we provide an overview of the UTXO sets of Bitcoin, Bitcoin Cash and Litecoin, the three UTXO-based cryptocurrencies with the highest market capitalization. Results included in this section correspond to the UTXO set as it was on 6 February 2018 and were made using the STATUS tool.

Unless the contrary is stated, all the plots included in this section show cumulative distribution functions. Therefore, a point $(x, y)$ in the plot shows the probability $y$ that a given variable (depicted in the $x$-axis label) will take a value less than or equal to $x$.

---

[2]It can be found under a bigger Bitcoin Tools library at https://github.com/sr-gi/bitcoin_tools/tree/v0.1/bitcoin_tools/analysis/status.

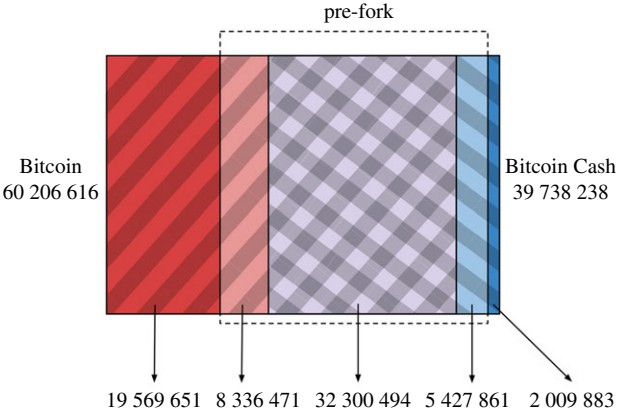

**Figure 1.** Venn diagram of the UTXO sets of Bitcoin and Bitcoin Cash.

**Table 1.** Summary of the main properties of the UTXO sets.

| coin | Bitcoin | Bitcoin Cash | Litecoin |
|---|---|---|---|
| UTXO set date | 6 Feb 2018 | 6 Feb 2018 | 6 Feb 2018 |
| block height | 507 964 | 516 120 | 1 364 009 |
| no. tx | 28 414 343 | 17 709 176 | 2 721 617 |
| no. UTXOs | 60 206 616 | 39 738 238 | 18 445 858 |
| avg. no. UTXOs per tx | 2.12 | 2.24 | 6.78 |
| s.d. no. UTXOs per tx | 16.96 | 26.10 | 80.64 |
| median no. UTXOs per tx | 1 | 1 | 1 |
| size of the (serialized) UTXO set | 3.44 GB | 2.27 GB | 1.04 GB |
| amount of coins | 16 849 386 BTC | 16 951 348 BCH | 55 098 258 LTC |

Bitcoin is the cryptocurrency with the highest market capitalization (135 thousand million dollars in April 2018).[3] Its source code was released in 2009, and since then many other cryptocurrencies have forked that code, sharing many details from the Bitcoin protocol into theirs.

Bitcoin Cash is a fork of the Bitcoin blockchain created on August 2017 as a consequence of the disagreements between the community about how to handle scaling problems in Bitcoin. Bitcoin and Bitcoin Cash share the same blockchain history up to block 478 558, where they forked and each coin followed a different path. On February 2018, Bitcoin and Bitcoin Cash still shared 32.3 million UTXOs (figure 1), which, respectively, constitute 53.65% and 81.28% of Bitcoin and Bitcoin Cash UTXO sets. Moreover, 40 636 965 of the UTXOs existing in Bitcoin in February 2018 belong to transactions that were made before the fork (67.50%); on the other side, 37 728 355 of the UTXOs existing in Bitcoin Cash were made before the fork (94.94%). This means that, at the time of analysing the data, Bitcoin Cash has consumed 8.3 million of UTXOs existing before the fork, and has created a smaller number of new ones (2 million), whereas Bitcoin has consumed 5.4 million outputs and created two orders of magnitude more new ones (19.5 million).

On the contrary, Litecoin is a coin whose chain is totally independent of the Bitcoin blockchain. However, the source code of Litecoin is a *software* fork[4] of the original Bitcoin Core client created in 2011. Litecoin mainly differs from Bitcoin in the average time between blocks (2.5 min instead of 10), the hashing algorithm (Scrypt instead of SHA256) and the increased maximum amount of coins (84 instead of 21 million).

Table 1 presents a summary of the main properties of the UTXO sets of Bitcoin, Bitcoin Cash and Litecoin, for the snapshot taken on 6 February 2018. The difference on the time between blocks in Litecoin versus both Bitcoin versions can be clearly appreciated by its block height, which was 1.3

---

[3]See https://web.archive.org/web/20180418064506/https://coinmarketcap.com/.

[4]A software fork as in a github fork, not to be confused with a soft/hard fork of the chain.

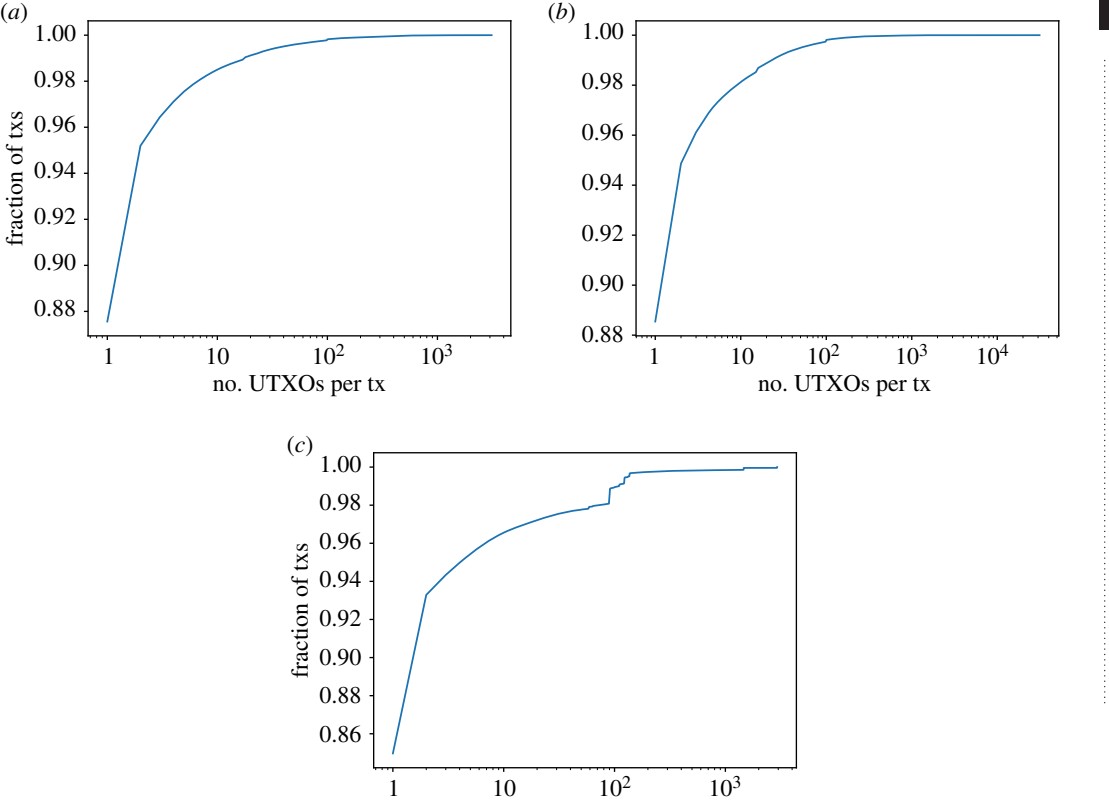

**Figure 2.** Number of UTXOs per transaction in (*a*) Bitcoin, (*b*) Bitcoin Cash and (*c*) Litecoin.

million on February 2018 even though the coin started 2 years after Bitcoin. By contrast, Bitcoin and Bitcoin Cash height were around half a million. Similarly, the changes on the coin supply are also clearly reflected in the UTXO set: Litecoin's current supply is already higher than the supply limit of Bitcoin. Litecoin also presents a significantly less amount of both UTXOs and transactions. As a result, the size of its UTXO set is smaller. However, the average number of UTXOs per transaction in Litecoin is three times bigger than Bitcoin's.

The distributions of the number of UTXOs per transaction are very skewed (figure 2). Even when Bitcoin has an average of 2.12 UTXOs per transaction (2.24 for Bitcoin Cash), most of the transactions have just one unspent output. 87.6% of transactions in Bitcoin and 88.5% in Bitcoin Cash have only 1 UTXO and 97.1% (96.8%) have less than 5 in Bitcoin (Bitcoin Cash). For Litecoin, 85% of the transactions have a single UTXO. It is worth noting that almost 1% of transactions in Litecoin have between 80 and 110 UTXOs (0.44% of them have exactly 90), a phenomenon not observed in either version of Bitcoin. The maximum number of UTXOs per transaction in the chosen snapshots is exhibited by Bitcoin Cash, with a transaction having 31 243 UTXOs[5] (from a total of 31 244 outputs).

Regarding the transaction height of the UTXOs in the sets (figure 3), some differences can be already appreciated in Bitcoin versus Bitcoin Cash: while Bitcoin shows a similar trend regarding UTXOs in the lasts blocks, there are almost no UTXOs from the firsts blocks after the fork in Bitcoin Cash, although the tendency seems to recover after a few blocks. Litecoin follows a different pattern, with as much as 67% of its UTXOs being from the first five months of the coin (first 100 683 blocks). By contrast, in Bitcoin, half of the stored UTXOs are older than December 2016 (block 441 704 corresponds to the median), whereas the other half are younger. Such a phenomenon shows how almost half of the current UTXO set is composed by UTXOs created in 2017 and 2018. On the other hand, there are still very old UTXOs: 2% of them are older than October 2012 (block height 201 408).

Figure 4 shows the cumulative distribution function of the heights of coinbase transactions with UTXOs. For Litecoin, the probability of a coinbase transaction with UTXO(s) being at any given height is almost the same. On the contrary, for both Bitcoin versions, it is more likely for such a transaction to belong to an old block than to a newer one (half of those transactions have heights

[5]See https://blockchair.com/bitcoin-cash/transaction/dbd3f7518111d679c1b229af71181c9395e3bf8c1370b6856376f391d25c883e.

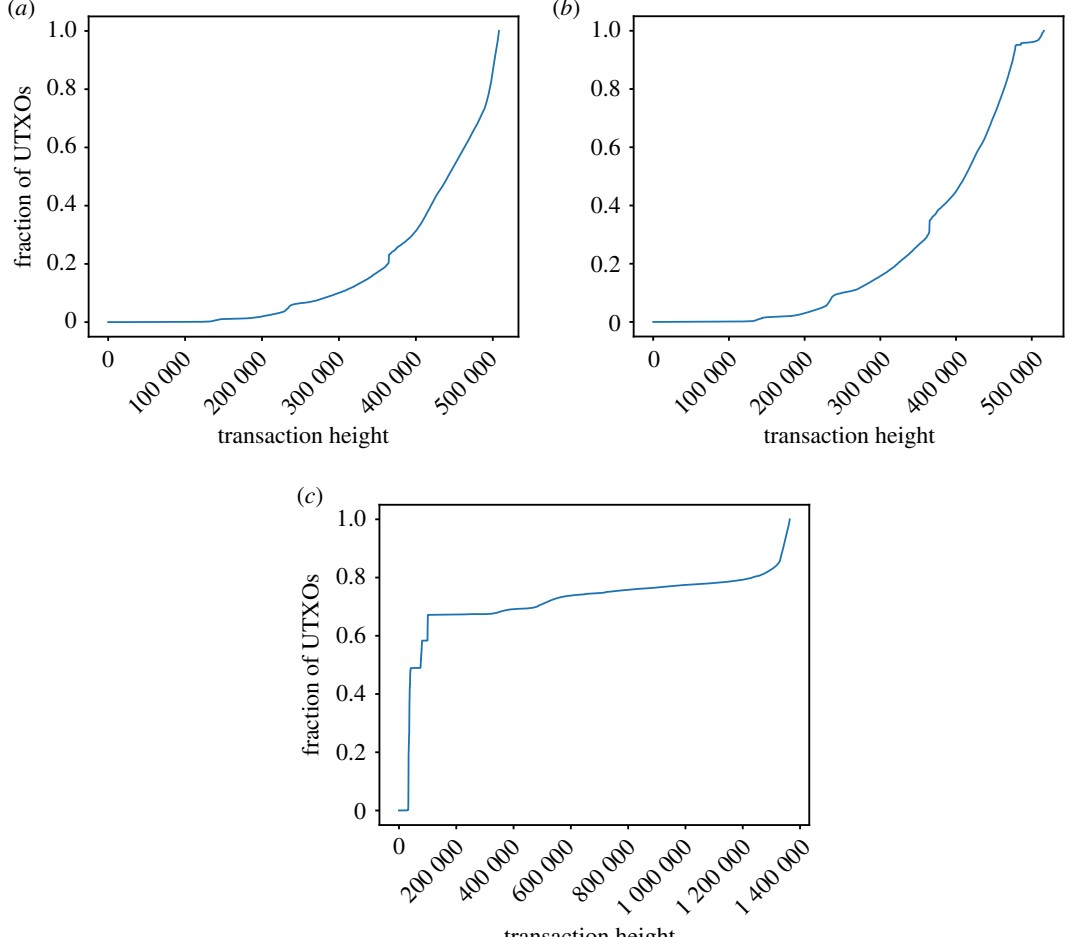

**Figure 3.** Number of UTXOs per height in (*a*) Bitcoin, (*b*) Bitcoin Cash and (*c*) Litecoin.

lower than 61 050 in Bitcoin, 84 897 in Bitcoin Cash). Note the contrast with the analysis considering all UTXOs, regardless of whether they are coinbase or not (figure 3). In Bitcoin, 75% of the coinbase outputs in the UTXO set were created before block 278 983 (January 2014). By contrast, just 8% of the current UTXOs were created before that block. Regarding the differences between Bitcoin and Bitcoin Cash, it is also worth noting that for Bitcoin Cash, 17.53% of the coinbase transactions with UTXO(s) were created after the fork, a phenomenon not observed in Bitcoin.

Bitcoin and Litecoin also present clear differences regarding the amounts stored in the UTXOs. 66.7% of the UTXOs in Litecoin store the smallest unit of value possible, that is, just 1 litoshi. By contrast, just 1.4% of Bitcoin's UTXOs store 1 satoshi. Moreover, the vast majority of UTXOs are smaller than one bitcoin/litecoin (98.8% and 94.7%, respectively). However, both currencies have one thing in common: the preference for output amounts that are powers of 10. In Bitcoin, the most common amount is $10^3$ satoshis (4% of the UTXOs), followed by $10^4$ and 1 (1.9% and 1.4%, respectively). This can be seen in the plots of figure 5 by the small upticks on the plotted line. In Litecoin, the most common amount is 1 (66.7% of the UTXOs), followed by $10^6$ and $10^8$ (0.7% and 0.5%, respectively).

The previous figures draw attention to the fact that there are a lot of UTXOs with an amount equal to 1. To further explore this phenomenon, figure 6 shows the height of UTXOs with such amount (i.e. 1 satoshi or litoshi). For Litecoin, 99% of the UTXOs with an amount equal to one have height 100 636 or lower, that is, they were made before 15 March 2012. It turns out that these UTXOs were created as an attack to Litecoin, and the community even considered marking these outputs as unspendable so that they could be removed from the UTXO set. The debate about how to handle these UTXOs is still alive while we are writing these lines.

Table 2 presents the amount of UTXOs divided by script type.[6] In all the analysed coins, P2PKH is the most common script type, representing 81% and 94% of the UTXOs in Bitcoin and Bitcoin Cash,

---

[6]See §4.2 for a detailed description of each UTXO type.

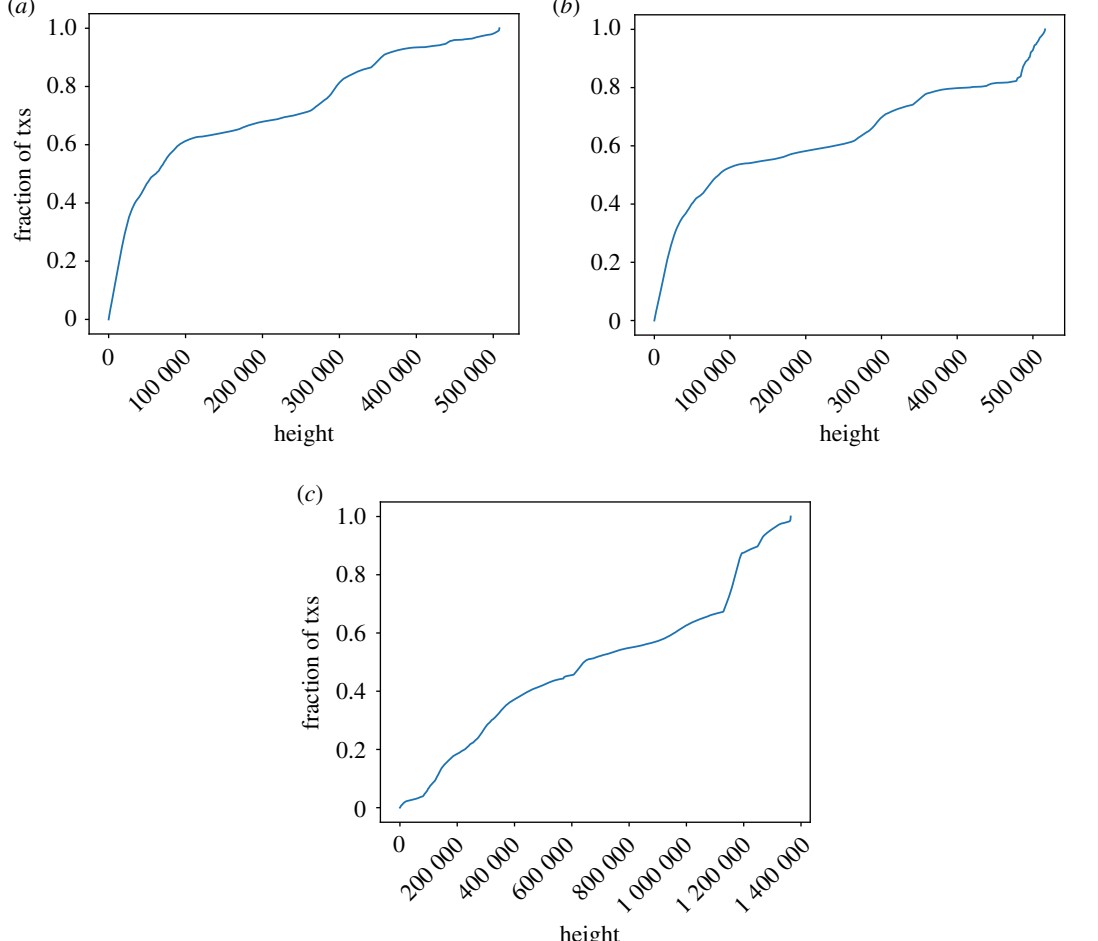

**Figure 4.** Transaction height of the UTXO of coinbase transactions. (*a*) Bitcoin, (*b*) Bitcoin Cash and (*c*) Litecoin.

respectively, and 97% of them in Litecoin. It is worth mentioning that there are no segregated witness scripts in Bitcoin Cash, since such kinds of scripts do not exist in the coin. Litecoin has a significant amount less of both multisignature scripts and P2PK scripts using compressed keys.

Figure 7 shows the evolution in time of the different types of outputs in the UTXO set. Regarding Bitcoin, most P2PK UTXOs were created during the very first years, which is an expected result since P2PKH was developed afterwards as an improvement of P2PK. However, it is interesting to see that, after a long time with very few outputs of this type, around March 2017 and during 324 blocks, 15% of the current P2PK outputs included in the UTXO set were created. There are no multisig, P2SH, P2WPKH nor P2WSH from the first blocks, since these kinds of scripts were created years after the creation of the coin.

Regarding Litecoin (figure 7*c*), it is surprising to see that all multisig UTXOs in the snapshot were created in a very short time: 99% of them were created before block 753 252 and 96% of them between blocks 743 428 and 753 252. All existing non-standard UTXOs were only created during the first 487 470 blocks. P2PKH and P2PK UTXOs were mostly created in the very beginning (as most of the coins).

# 4. Dust and unprofitable UTXOs

An interesting type of output included in the UTXO set is that whose economic value is small enough to pose a problem when it has to be spent. Since such UTXOs are in the set, they occupy space and thus convey a cost to the system. However, the cost of spending them is so high with respect to its value, that the owner has no direct economic incentive to do so.

Bitcoin Core currently [6] defines a *dust* output as an output that costs more in fees to spend than the value of the output. To compute the cost of spending an output, both its size and the size of the input from the transaction that will spend it, are required. Since the input that spends a given UTXO is not yet in the blockchain Bitcoin Core sets such value to 148 regardless of the type of outputs that will generate

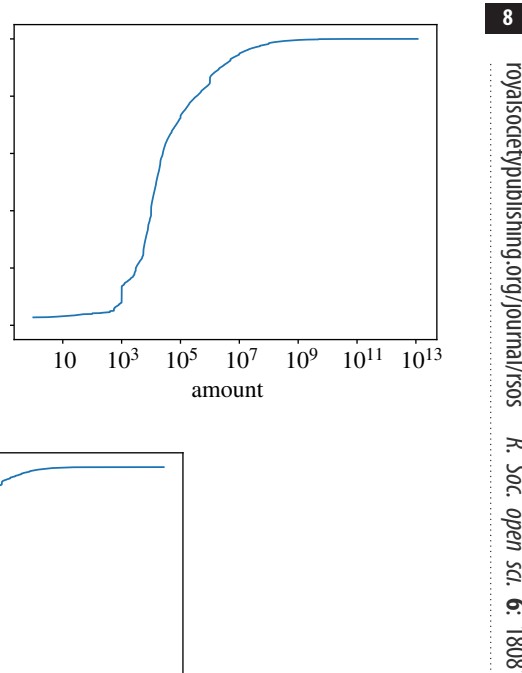

**Figure 5.** Amount in UTXOs. (*a*) Bitcoin, (*b*) Bitcoin Cash and (*c*) Litecoin.

the input. These 148 bytes are based on the most common type of Bitcoin outputs, P2PKH and account for the outpoint $(32 + 4)$, the length of the script in bytes (1), the signature $(1 + 72)$, the public key $(1 + 33)$ and the sequence number (4).

Therefore, a given UTXO out is considered dust by the Bitcoin Core client with respect to a fee-per-byte rate $\mathrm{f}$ with the following definition:

$$\text{is\_dust(out)} = \begin{cases} 1, & \text{out}_v < \mathrm{f} * (41 + 107/\alpha + \text{out}_s) \\ 0, & \text{otherwise,} \end{cases}$$

where $\text{out}_v$ is the amount deposited in the output; $\text{out}_s$ is the size (in bytes) of the output; $\mathrm{f}$ is the current fee rate (in satoshis per byte) and $\alpha$ is the segwit discount factor. If the output is not a native segwit script, then $\alpha$ is 1. However, if it is indeed a segwit output, the size of the input script needed to redeem it is discounted by a factor $\alpha = 4$.

The definition of dust, as interpreted by the Bitcoin Core, has two main limitations. First, it assumes a fixed transaction input size of 148. Although the exact content of the input of a UTXO will never be, by definition, known to the client, in fact, the size of an input can usually be predicted with high accuracy. Second, it considers both the sizes of the output and the input. Although this may be useful in some cases, we claim that since the transaction containing the output is already in the blockchain, its size should not be taken into account when analysing the dust problem (since it has already been paid).

Therefore, we define an *unprofitable* output as the output of a transaction that holds less value than the fee necessary to be spent, taking into account only the size of the input that will be needed to spend it.

$$\text{is\_unprofitable(out)} = \begin{cases} 1, & \text{out}_v < \mathrm{f} * \text{pred\_in}_s \\ 0, & \text{otherwise,} \end{cases}$$

where $\text{pred\_in}_s$ is the predicted size of the input that will spend output out. That is, instead of using a predefined 148-byte size, we compute as accurately as possible the size of the input that is going to spend the output.

However, it is not always possible to exactly determine the size of the input of a certain UTXO. Depending on the type of the script found in the UTXO, we will be able to accurately determine the

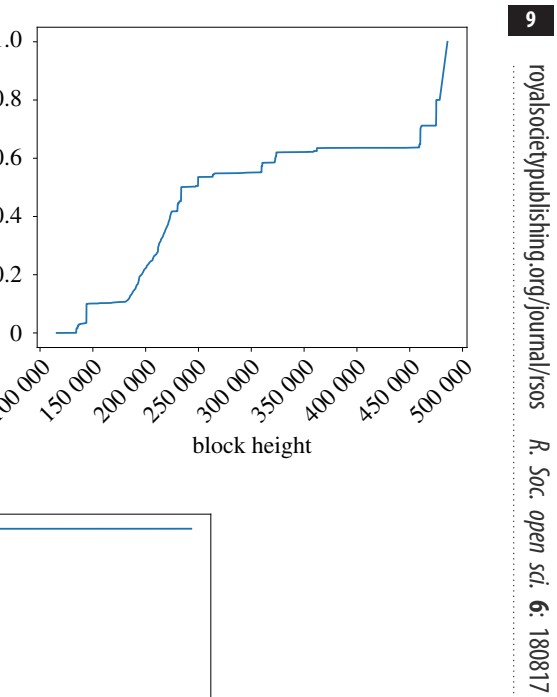

**Figure 6.** Transaction height of the UTXOs of amount 1 satoshi/litoshi. (*a*) Bitcoin, (*b*) Bitcoin Cash and (*c*) Litecoin.

**Table 2.** UTXO types.

|  | Bitcoin | Bitcoin Cash | Litecoin |
|---|---|---|---|
| no. UTXOs | 60 206 616 | 39 738 238 | 18 445 858 |
| Pay-to-PubkeyHash (P2PKH) | 48 884 862 | 37 457 455 | 17 949 409 |
| Pay-to-ScriptHash (P2SH) | 10 825 371 | 1 825 348 | 399 987 |
| Pay-to-Pubkey (P2PK) | 75 394 | 77 311 | 81 399 |
| compressed | 39 479 | 40 956 | 457 |
| uncompressed | 35 915 | 36 355 | 80 942 |
| Multisig | 408 847 | 377 355 | 325 |
| P2WPKH | 6 176 | — | 76 |
| P2WSH | 5 188 | — | 14 |
| others (non-standard) | 778 | 769 | 14 648 |

size that its input will have, or we will have some uncertainty about it. Therefore, we will consider two different metrics for unprofitability: a lower bound on unprofitability (is_unprofitable$_{low}$), that will take into account the minimum size of the input; and an estimation of unprofitability (is_unprofitable$_{est}$), that tries to estimate as accurately as possible the real unprofitable rates taking into account data available in the blockchain.

In order to identify the three types of outputs (that is, dust, unprofitable$_{low}$ and unprofitable$_{est}$), it is important to recall that the amount of fees a transaction has to pay to be included in a new block depend

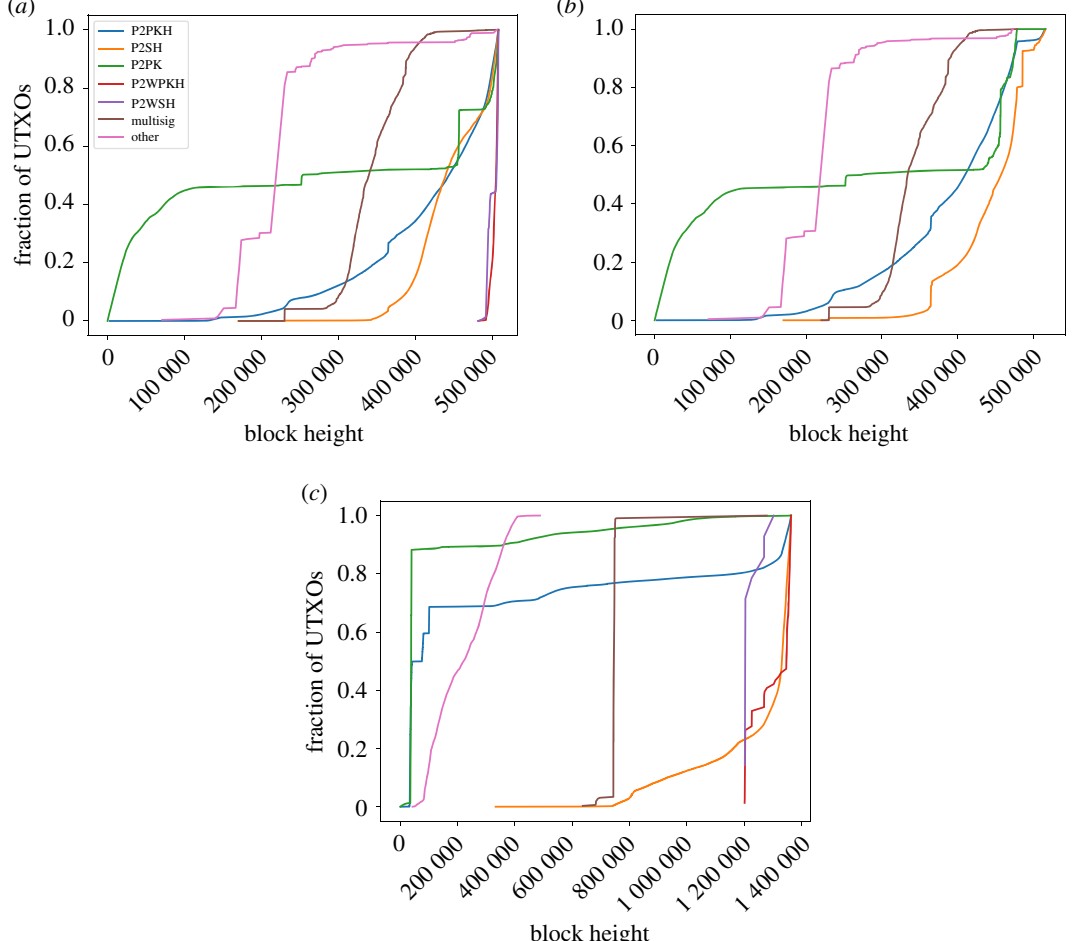

**Figure 7.** UTXO output type per height in (*a*) Bitcoin, (*b*) Bitcoin Cash and (*c*) Litecoin.

on two factors: the *fee-per-byte rate*, ƒ, that the network is expecting at the time of creating the transaction and the *size* of the transaction. Section 4.1 explains what is the fee-per-byte rate and shows its value over the past years. Regarding the size of the transaction, the definition of dust uses a fixed size, but our unprofitability approach takes into account the minimum size of the input. Section 4.2 explains what are the minimum inputs of UTXOs depending on their type. Finally, the estimation of unprofitability needs to take into account the sizes of variable parameters. Section 4.3 presents the estimations of those sizes made from blockchain data.

## 4.1. Fee-per-byte rate

Block space is a scarce resource. Whenever the throughput of transactions sent to the P2P network is higher than what blocks can accommodate, transactions pending to be included in blocks start to accumulate in the mempool. When a block has to be created, miners choose the set of transactions to include in the block from those stored in the mempool. Assuming miners are rational actors and given that block space is limited, transactions paying a higher fee-per-byte rate should be more attractive to miners, since such transactions would usually provide the highest revenue. However, there are many both technical and non-technical factors that may affect the miners' decision in choosing a certain transaction for inclusion in a block, e.g. the number of signature check operations it contains, the cost of recomputing the merkle tree in order to include a new transaction, or even the attempt to censor a certain transactions.

The fee-per-byte rate measured in satoshis (or litoshis, for litecoin) per byte is a highly variable factor that depends on the transaction backlog (i.e. how many transactions are pending to be included in new blocks and how much are those transactions willing to pay to be included in a block).

Figure 8 shows the average fee per byte paid for transactions included in blocks, from the creation of the coin up until nowadays. Bitcoin and Bitcoin Cash show differences on the blocks after the fork, with

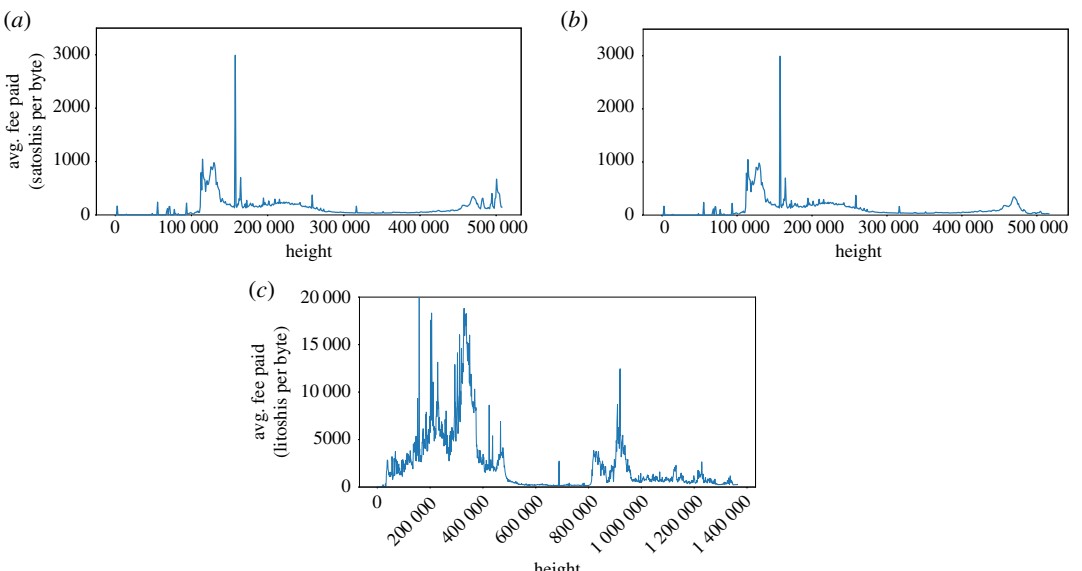

**Figure 8.** Average fee rate (in satoshis/litoshis per byte) paid by transactions at a given block height. The plot is showing 1000 block averages. (*a*) Bitcoin, (*b*) Bitcoin Cash and (*c*) Litecoin.

Bitcoin Cash presenting lower fee rates. Litecoin presents higher variability. In Bitcoin and Bitcoin Cash, the highest peak is produced by block 157 138, that contains a transaction[7] that paid as much as 85.9 BTC in fees. In Litecoin, the highest peak is found in block 158 277, that has a single non-coinbase transaction[8] paying 100 LTC as fees.

## 4.2. Determining the size of the minimum-input of an UTXO

Unprofitability is defined considering the size of the minimum-input of each UTXO. In order to label the outputs in the UTXO set as unprofitable, we need an estimation of the size of data required to spend such output. To identify such minimum information, we can consider an already standard transaction with its inputs and its outputs and enough fees to be relayed. Then, we define the *minimum-input of a UTXO* as the smallest size input that spends such UTXO. The size of such minimum-input, along with the value held in the output and the fee rate, will determine whether a UTXO may be flagged as unprofitable.

To measure the size of such minimum-input, we need to review the structure of a Bitcoin transaction. As depicted in figure 9, all transactions follow a standard structure containing some fixed length parameters that determine a minimum transaction size, and some variable length parameters, depending on the transaction type. When a transaction is created, inputs are defined referring to some UTXOs. Such inputs have different sizes depending on the output type they are related to. On the other hand, new outputs are generated for every new transaction, and thereby some additional size, which will depend on the new output type, will be added to the transaction.

Depending on the UTXO type, its minimum-input size will be different. Such measure can be split in two parts: fixed size and variable size. Regarding the fixed size, as depicted in figure 9 (taking into account only the input box), we can identify three fields: `prev_tx_id`, `pev_out_index` and `nSequence`. Therefore, for every UTXO, its minimum-input will be at least 40 bytes long independently of its type. On the other hand, the content and length of the fields `scriptSig` and `scriptSig length` depend on the UTXO type, specified in the field `scriptPubKey` of the UTXO.

The different types of outputs, with their corresponding size, can be classified as follows:

*Pay-to-PubKey (P2PK) outputs.* The minimum-input of this type of UTXO specifies just a digital signature to redeem the output and the `scriptSig` includes the following data:

 `PUSH sig (1 byte)+sig (71 bytes)`

Bitcoin uses DER encoded ECDSA signatures in the scripts of its transactions, which can be between 71 and 73 bytes long depending on their `r` and `s` components [7]. Such variability comes from the

---

[7]See https://blockchain.info/tx/1dbd420f23f8ac58b994c7b84f8667670dfcad39ec9f34e555a0c6281b7e2c04.

[8]See https://chainz.cryptoid.info/ltc/block.dws?2465020d67d0f08f53d1df0571af8aadd81b7b60ed27e2f236e1b08e6321547f.htm.

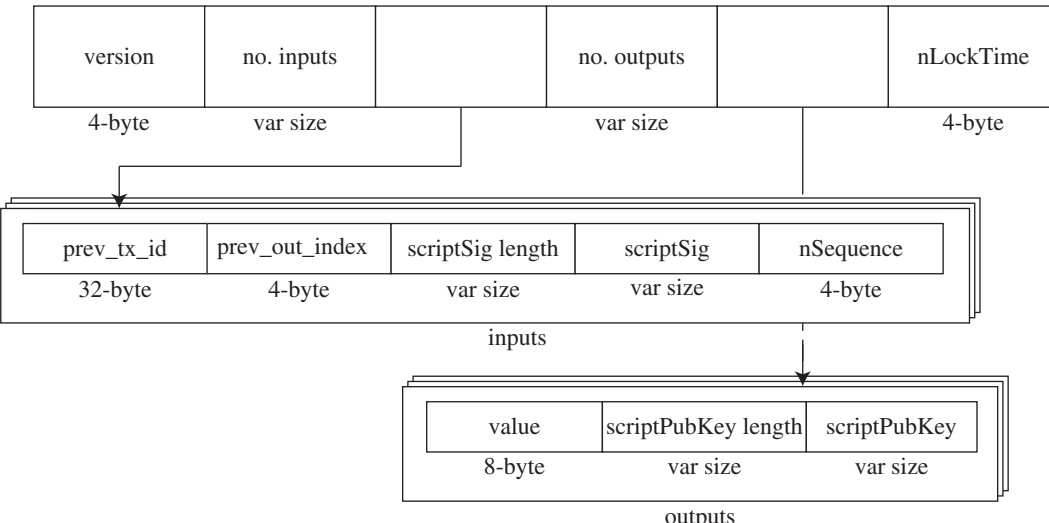

**Figure 9.** Generic transaction structure.

randomness of the r parameter. Since we are defining the minimum possible input to be created, 71-byte signatures are considered. Hence, the `scriptSig` for a P2PK UTXO will be 72 bytes long and `scriptSig len` field will be 1 byte long, resulting in a minimum-input size of 73 bytes.

*Pay-to-PubkeyHash (P2PKH) outputs.* For this UTXO to be redeemed, both a signature (*sig*) and a public key (*pk*) are required in the `scriptSig`, as shown below:

```
PUSH sig (1 byte)+sig (71 bytes)+PUSH pk (1 byte)+pk (33-65 bytes)
```
Regarding the signatures, the same assumption as for P2PK outputs applies, that is, 71-byte length can be considered. Regarding public keys used by Bitcoin, they can either be compressed or uncompressed, which will significantly vary their size:

— Uncompressed keys: such keys were used, by default, in the first versions of the Bitcoin Core client, and they are 65 bytes long.
— Compressed keys: in 2012, Bitcoin Core started using this more efficient type of keys, which are almost half size of the previous ones (33 bytes), and therefore make smaller scripts.

So, the size for the `scriptSig` varies from 106 to 138 and therefore, the `scriptSig length` field will be 1 byte long, resulting in a total minimum-input size between 107 and 139 bytes.

*Pay-to-multisig (P2MS) outputs.* The size of the minimum-input to redeem such a script varies highly depending on the number of signatures required, which ranges up to 20 (20-of-20 multisig),[9] so the `scriptSig` for redeeming such output is as follows:

```
OP_0 (1 byte)+(PUSH sig (1 byte)+sig (71 bytes)) *
required_signatures (1-20)
```
Thus, the size of the `scriptSig` field will range between 73 and 1441 bytes, making the `scriptSig len` field range between 1 and 2 bytes, so the total minimum-input size will be between 74 and 1443.

*Pay-to-ScriptHash (P2SH) outputs.* Unlike any previous output type, input size created from P2SH outputs cannot be straightforwardly defined in advance. P2SH outputs hide the actual input script behind a hash, to make smarter outputs, by making them smaller and thus allowing the payer to pay lower fees. However, the scripts held by those UTXOs give us no clue about how the minimum-input should be build. Therefore, we have set the minimum-input size to just the fixed size (40 bytes) plus an additional byte for the script length, resulting in 41 bytes.

Native *Pay-to-Witness-Public-Key-Hash (P2WPKH)* and *Pay-to-Witness-Script-Hash (P2WSH) outputs.* These new types of outputs are redeemed with empty input scripts. Data that prove ownership are moved into the witness and get a discount when computing the size of the input. Therefore,

[9]Although the standard considers a maximum number of three signatures in a P2MS output, up to 20 are valid regarding the consensus rule [8] so they could potentially be found in the UTXO set.

**Table 3.** Minimum-input size summary.

| UTXO type | fixed fields | scriptSig. len. | scriptSig/redeemScript | | | | |
|---|---|---|---|---|---|---|---|
| | | | sig size | pk size | push data | total size |
| P2PK | 40 | 1 | 71 | — | 1 | 113 |
| P2PKH | 40 | 1 | 71 | 33 − 65 | 2 | 147 − 179 |
| P2MS | 40 | 1 − 2 | 71 * m (m ∈ [1, 20]) | — | m + 1 | 42 + 1/2 + 72*m |
| P2SH | 40 | 1 − var | var | var | var | 41 − var |
| P2WPKH | 40 | 1 | 71 | 33 | 2 | 68 |
| P2WSH | 40 | 1 | var | var | var | 41 − 10041 |

native segwit outputs will need the 40 fixed length bytes, the script size (1) and the discounted witness size.

The witness script needed to redeem a P2WPKH script is

```
PUSH sig (1 byte)+sig (71 bytes)+PUSH pk (1 byte)+pk (33 bytes)
```

Note that, in contrast with P2PKH scripts, here the public keys will always be found in a compressed form, as only this format is accepted in P2WPKH scripts. Therefore, P2PKH scripts will always be 106 bytes, and their inputs will need $40 + 1 + 106/4 = 68$ bytes

The witness script to redeem a P2WSH output is equivalent to the input script of a P2SH output. However, P2WSH are not affected by the 520-byte push limit of P2SH scripts.

Table 3 summarizes the sizes of the minimum-input for each UTXO type, where var indicates the values are variable and cannot be determined in advance.

## 4.3. Estimating variable values

As introduced in the last section, the inputs of some of the output script types contain values whose size cannot be known before actually seeing the input. However, the moment an output of such type is spent, we can observe the content of the corresponding input script and, therefore, compute its size. Yet whenever the output is spent, it is no longer an Unspent Transaction Output (UTXO), and it is thus no longer the subject of our analysis. On the other hand, we can indeed use historic data about outputs that have already been spent to estimate the input sizes of unspent outputs of the same type.

In this section, we provide an estimation of the variable-size parameters of input scripts based on the real data that already appears in the Bitcoin blockchain. Specifically, we analyse the sizes of public keys in P2PKH inputs, the sizes of P2SH inputs, the sizes of non-standard inputs and the sizes of P2WSH witnesses.[10] The estimation has also been performed for both Bitcoin Cash and Litecoin and the obtained values have been used in the analysis performed in §5. However, only the analysis for the Bitcoin blockchain will be broken down into parts due to space limitations.

*Public keys* can be found either in a compressed or uncompressed format. Public keys may be found both in input and output scripts: they may appear in P2PK and multisig outputs or in inputs spending P2PKH outputs. In our analysis, we have focused on checking public keys in inputs spending P2PKH outputs, since these are the ones that will affect our profitability analysis. The first compressed public key in a P2PKH input can be found in block 158 216.[11] The input spends an output in block 158 214.[12]

The use of compressed or uncompressed public keys highly depends on the height of the block: whereas there were no compressed public keys in the beginning, nowadays its use is prevalent. For this reason, we are interested in knowing how likely it is for a P2PKH output at a given height to be

[10]The raw data to perform such an estimation has been obtained from BlockSci and http://blockchain.info, and processed and included within STATUS.

[11]In the input of transaction https://blockchain.info/tx/94af4607627535f9b2968bd1fbbf67be101971d682023d6a3b64d8caeb448870.

[12]See https://blockchain.info/tx-index/2576077/0.

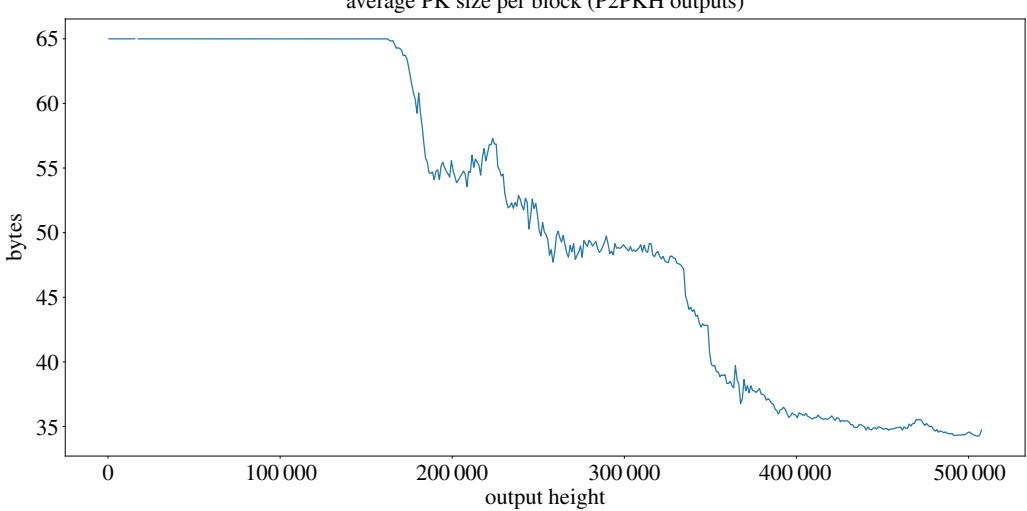

**Figure 10.** Block average public key sizes for P2PKH outputs. The plot is showing 1000 block averages.

spent by an input with a compressed or uncompressed public key. Figure 10 shows, for P2PKH outputs at a given height, the average public key size of inputs spending them. The plot is generated by analysing the 654 245 749 public keys found in P2PKH inputs in the blockchain, resulting in an overall average public key size of 39.3 bytes. Before block 158 214, all spent P2PKH outputs have uncompressed public keys in the corresponding inputs, because this was the format being used by Bitcoin clients at that time. Consequently, the plot shows a smooth line on the low heights, denoting the constant 64-byte length of public keys being used. The shift towards using compressed public keys is progressive, so the average public key size decreases from that point until block 450 000, when it stabilizes at around 34.5 bytes. Such behaviour indicates that, despite the obvious advantage of using compressed public keys, some Bitcoin users are still using uncompressed keys.

*P2SH inputs* must contain the redeem script (whose hash matches the one specified in the UTXO) and any data that are required to make the redeem script evaluate to *true*. Both items must be taken into account when computing the size of the input.

Table 4 breaks down the P2SH redeem scripts by type found in the blockchain, describing the average and the standard deviation of the length of their input script. That is, they consider the size of both the redeem script and the data consumed by that script.

Multisig scripts are by far the most common script encapsulated within P2SH, and they account for 85.6% of the P2SH redeem scripts in the blockchain. Because native multisig outputs are costly, it is common to encapsulate them in a P2SH, transferring the fees to the redeemer of the output. The size of a P2SH multisig input is determined by the number of allowed signers (that determines the number of public keys) and the number of required signers (that determines the number of signatures). Although a wide variety of encapsulated multisig scripts can be found in the blockchain, 2-of-2 and 2-of-3 multisig account for 97, 85% of multisig scripts (table 5 shows the amount of multisig scripts of each kind found in the blockchain).

The activation of segwit added four more types of output scripts to the existing set: two native scripts and the two P2SH nested versions. The input of a P2WPKH script nested in a P2SH is always 23 bytes long, whereas a P2WSH nested in P2SH is always 35. This accounts for the hash (20 and 32 for P2WPKH and P2WSH, respectively), plus the marker ($0 \times 00$) and two data pushes.

Non-standard redeem scripts may be of arbitrary length. However, the redeem script has to be pushed to the stack, so it is affected by the maximum stack element size of 520 bytes.[13] The average size of P2SH scripts with non-standard redeem scripts is 169.98 bytes, although most of them (66.06%) are just 11 or 12 bytes. The largest input script of this type is 9 319 bytes long.

Inputs of P2SH scripts with nested P2PK scripts may be either 108 or 140 bytes long (assuming 71-byte signatures), depending on whether they use compressed or uncompressed public keys. Nevertheless, data show that almost all of them have compressed public keys (there are just six of such scripts in the blockchain using uncompressed public keys). Similarly, inputs of P2SH

[13]See https://github.com/bitcoin/bitcoin/blob/0277173b1defb63216d40a8d8805ae6d5d563c26/src/script/script.h#L23.

**Table 4.** Summary of P2SH redeem scripts in the blockchain.

| redeem script | number of inputs | average input size | std. input size |
|---|---|---|---|
| Multisig | 80 839 329 | 241.6 | 22.7 |
| P2WPKH | 7 961 073 | 23 | 0 |
| P2WSH | 5 544 793 | 35 | 0 |
| Nonstd | 112 354 | 169.98 | 333.59 |
| P2PK | 23 557 | 108.01 | 0.51 |
| P2PKH | 448 | 132 | 0 |
| P2SH (hash puzzle) | 82 | 28.73 | 23.75 |
| total | 94 481 636 | 210.93 | 78.60 |

**Table 5.** Summary of multisig configurations found in P2SH redeem scripts in the blockchain.

| multisig specs. | number of scripts found | multisig specs. | number of scripts found |
|---|---|---|---|
| (2 of 3) | 56 498 831 | (1 of 3) | 21 334 |
| (2 of 2) | 22 600 200 | (3 of 6) | 17 131 |
| (2 of 6) | 491 146 | (3 of 3) | 13 357 |
| (3 of 4) | 453 256 | (4 of 5) | 10 456 |
| (1 of 1) | 258 449 | (2 of 5) | 8 826 |
| (2 of 4) | 209 809 | (4 of 6) | 7 918 |
| (3 of 5) | 144 958 | (1 of 6) | 5 864 |
| (1 of 2) | 86 678 | others | 11 116 |

scripts with nested P2PKH scripts may be either 132 or 164 bytes long, although all of them used compressed keys.

Finally, there are 82 redeem scripts that contain hash puzzles, that is, they ask for the preimage of a hash. We have found there are just two different versions of these scripts, one asking for the preimage of `OP_0` and the other asking for the preimage of a 152-byte length value. These make input sizes of 25 and 178 bytes, respectively.

In contrast with compressed and uncompressed public key usage, whose behaviour highly depends on the block height, we have found a small correlation between P2SH input script sizes and the block height where they are found (figure 11). There are no blocks with P2SH input scripts up until block 170 060 and the script sizes start to decrease at height 480 000. From block $\approx$ 275 000 up until block $\approx$ 480 000, the average size of inputs is almost constant, because 98.2% of them are 2-of-2 and 2-of-3 multisig scripts which are 216 and 250 bytes long, respectively. However, after this period, P2WSH and P2WPKH scripts encapsulated in P2SH inputs become popular, and because its size is smaller, the average P2SH input script size decreases.

Inputs spending *non-standard* outputs may contain scripts of arbitrary length. Since such inputs do not have a predefined structure, we resort to analysing the sizes of inputs spending non-standard outputs in the blockchain. We have found 220 758 of such inputs in the blockchain, which correspond to 78 different input scripts. The average size of non-standard input scripts is 1.04 bytes (99.3% of these inputs scripts are a single byte long, and 99.94% have less than three bytes). As a matter of curiosity, the input script `OP_TRUE` is found in 219 191 inputs.

These 220 758 non-standard inputs are found in just 143 different blocks, so data is too disperse to evaluate its size with respect to block height.

As in P2SH inputs, P2WSH inputs must also include the redeem script with a matching hash and any data needed to redeem it. However, instead of including the redeem script in the input itself, the redeem script is found in the witness and thus gets discounted by a factor $\alpha = 4$.

We have found 215 742 P2WSH inputs with an average witness size of 251.6 bytes. From block $\approx$ 488 000 onwards, the plot shows an stable trend. Of the redeem scripts, 97% are 252, 253 or 254 bytes

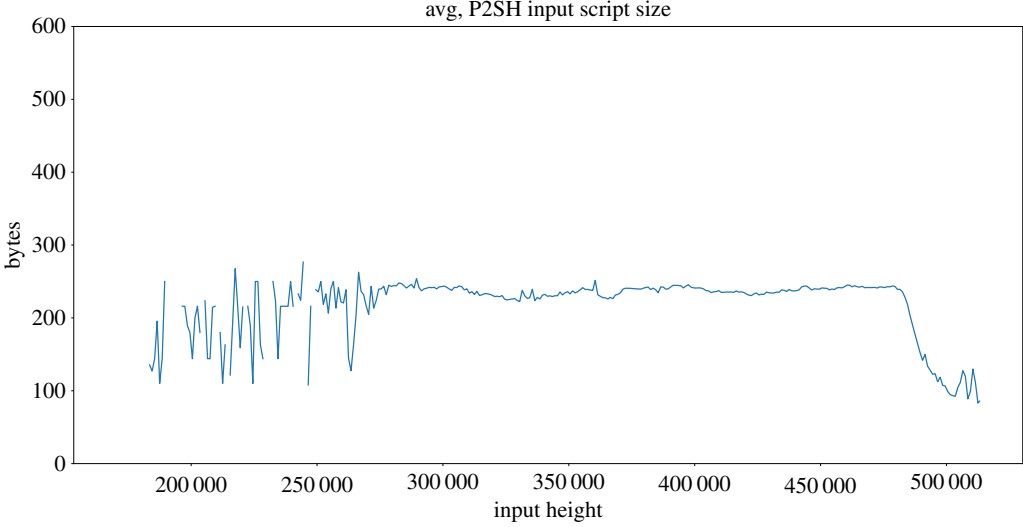

**Figure 11.** Block average P2SH input sizes. The plot is showing 1000 block averages.

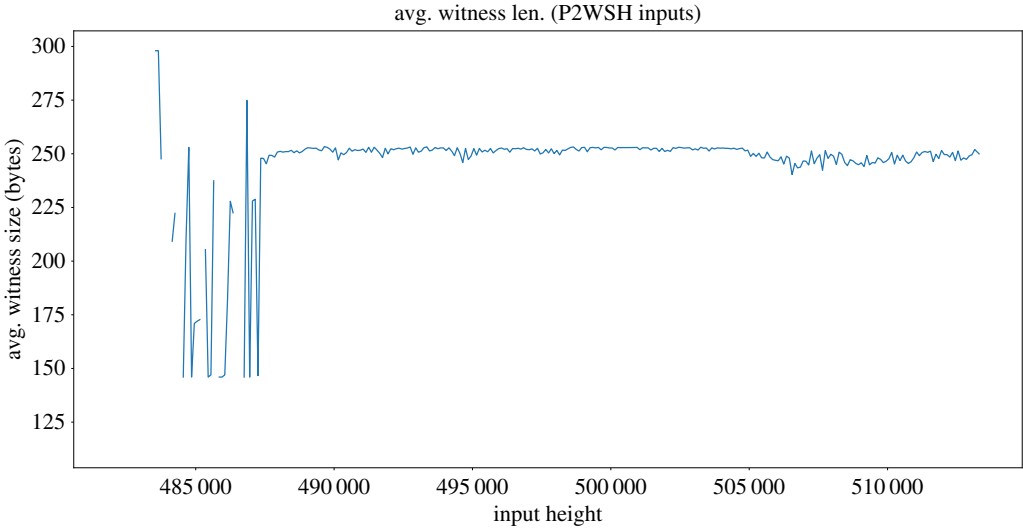

**Figure 12.** P2WSH inputs in the Bitcoin blockchain. The plot is showing 100 block averages. Average P2WSH witness size (bytes).

long, and correspond to 2-of-3 multisignature scripts. As P2SH input scripts, witness length is not clearly correlated with block height (figure 12). The first P2WSH input is found in block 482 133.

# 5. An analysis of dust and unprofitable outputs

In previous sections, we have seen that the two parameters that affect the amount of fees needed to spend a transaction are the size of the transaction and the current fee-per-byte rate. In this section, we first evaluate the current status of the UTXO sets of Bitcoin, Bitcoin Cash and Litecoin regarding the parameters previously defined: dust, a minimum bound on unprofitability (`unprofitable_low`) and estimation of unprofitability (`unprofitable_est`). After that, we analyse how unprofitability has evolved over time.

To study the impact of dust, we have considered the fixed input size of 148 bytes as defined by the Bitcoin client (discounting segwit when needed). For a `unprofitable_low`, we have considered the minimum size of the inputs. Whenever there is a parameter with variable length (whose size cannot be precisely known in advance), we have chosen the smallest possible value (recall §4.2). On the contrary, for `unprofitable_est` we have considered exact input sizes when known and resorted to estimations based on blockchain data when a variable-length parameter is needed, as described in

**Table 6.** Summary of script size estimations per script type and coin (computed using data in the blockchain).

| script type | estimation method | Bitcoin | Bitcoin Cash | Litecoin |
|---|---|---|---|---|
| P2PKH | block height average | [148, 180] | [148, 180] | [148, 180] |
| P2SH | absolute average | 210.93 | 240.99 | 184.06 |
| non-std | absolute average | 1.04 | 1.04 | — |
| P2WSH | absolute average | 251.5 | — | 217.5 |

§4.3. As explained in the previous section, the use of compressed or uncompressed public keys was highly dependant on the block height. On the contrary, P2SH, non-standard and P2WSH sizes do not show that tendency (compare figure 10 with figures 11 and 12). Therefore, to estimate variable sizes, we use the absolute average of the sizes found in the blockchain for P2SH, non-standard and P2WSH inputs, and the average for outputs at a given height for P2PKH. Table 6 presents a summary of the values used for the estimations. Note that P2PKH input scripts are always estimated to be between 148 and 180 bytes regardless of the analysed coin, although the exact distribution differs and is computed individually from the history of each coin.

As depicted in figure 8, fee-per-byte rate is far from fixed and has high variability. Thus, to take into account different possible scenarios, we have considered a broad fee-per-byte spectrum, ranging from 0 to 349 satoshi/byte.

Figure 13a shows the fraction of UTXOs that are considered dust, unprofitable$_{low}$ and unprofitable$_{est}$ in the Bitcoin UTXO set (blue, orange and green lines, respectively). For a fee-per-byte as small as 116 satoshi/byte onwards, more than the 50% of UTXOs (30.2 million outputs) from the set can be considered dust, whereas the same 50% size for unprofitable_low is reached for 232 satoshi/byte onward. Note that 5 617 outputs will always be considered dust and unprofitable, regardless of the fee rate, because they have an amount of 0.

Dust and the unprofitable_est present a very similar distribution, whereas the unprofitable_low always shows, as expected, a lower amount of unprofitable outputs. However, the similar trend shown by dust and unprofitable_est may lead to misinterpretation: although the overall distribution is indeed similar, the individual estimations are very different, but overestimations in one kind of samples compensate for underestimations of the other kind. For P2PKH outputs (which represent 81% of the set), dust always considers size of 182 bytes (148 of the input and 34 of the output). On the contrary, unprofitability will only take into account the size of the input. For the lower bound, it will be either 147 or 179 (depending if a compressed or uncompressed public key is considered, a decision taken on the basis of the block height). For the estimation, it will be any value between 148 and 180 and will depend on the average public key size found for outputs of the same height. Therefore, on one hand, dust will always estimate higher sizes than both unprofitability metrics and, on the other hand, the two unprofitability distributions are very similar, with the estimation offering slightly higher sizes than the minimum (recall figure 10). However, for P2SH scripts (which represent 18% of the set), the contrary is observed: dust predicts a fixed 180 bytes, whereas unprofitable_est predicts around 252 bytes and unprofitable_low predicts 42 bytes. As a consequence, in this case unprofitable_est always predicts much higher sizes. So although dust and unprofitable_est present similar distributions, it is important to note that size estimations for individual samples differ.

Figure 13b shows the relative size of dust and unprofitable output sets within the total UTXO set. Dust and unprofitable UTXOs represent a relevant part of the total size from the set (more than the 50% are dust for around 111 satoshi/byte onward), while the same can be seen for unprofitable_low UTXO for a rate of 204 satoshi/byte onward.

Finally, from an economic point of view, figure 13c shows, as expected, how those dust and unprofitable UTXOs represent a negligible amount from the total value of the UTXO set, that is, the total number of bitcoins in circulation.

Figure 14a shows the fraction of UTXOs that are considered dust, unprofitable$_{low}$ and unprofitable$_{est}$ in the Litecoin UTXO set. It is worth noting that 67% of the UTXOs are considered dust and unprofitable for a fee-per-byte as small as 1 litoshi/byte. Those UTXOs have output amounts of just 1 litoshi (recall figure 5c). Therefore, the three metrics (dust and the two unprofitability versions) consider all those UTXOs unprofitable at the same fee rate. From 300 litoshi/byte onwards, the same scenario

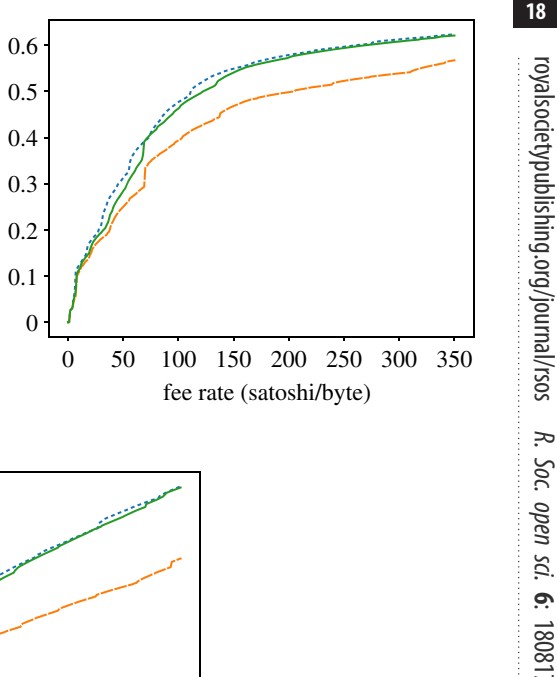

**Figure 13.** Dust and unprofitable analysis for Bitcoin. (*a*) Fraction of dust/unprof. UTXOs w.r.t. fee-per-byte rate, (*b*) fraction of occupied space w.r.t. fee-per-byte rate and (*c*) fraction of economic value w.r.t. fee-per-byte rate.

previously seen in Bitcoin is shown: dust estimates bigger sizes for P2PKH, whereas `unprofitable_est` estimates bigger sizes for P2SH outputs.

Finally, Bitcoin Cash dust and unprofitability analysis (figure 15) shows little difference with respect to Bitcoin, with dust and `unprofitable_est` showing very similar distributions although individual samples' estimations do differ.

In conclusion, we have seen that the three analysed coins exhibit lots of UTXOs that can be considered dust/non-profitable, and they account for an important part of the UTXO set in terms of size. However, their value is negligible.

## 5.1. Evolution of unprofitability over time

We have seen how for the most recent analysed UTXO set snapshot, namely data from 6 February 2018, the percentage of data that fall into any of the three categories (dust, unprofitable$_{low}$ and unprofitable$_{est}$) is far from being negligible. In this regard, we have analysed how such data have evolved since the early days of Bitcoin. To perform such analysis, we have focused on the unprofitable$_{est}$ metric since we think it is the most accurate one.

For Bitcoin, data have been obtained from nine snapshots from different heights of the blockchain, starting from block 100 000[14] and stepping every 50 000 blocks until height 500 000. Figure 16 shows the number of UTXOs in the set in Bitcoin over time (dots indicate the snapshots used in our analysis; table 7).

The percentage of unprofitable outputs per snapshot is almost always increasing at any point in the chart, as we can see depicted in figure 17*a*. However, there are a few exceptions, the 400K snapshot with respect to 450K and 500K being the most relevant. As we can see, the amount of unprofitable outputs located in the lower stages of the fee rate for the 400 K snapshot (between 0 and 75 satoshi/byte)

---

[14]We have not performed the analysis starting from block 1 due to the lack of relevant information in those snapshots.

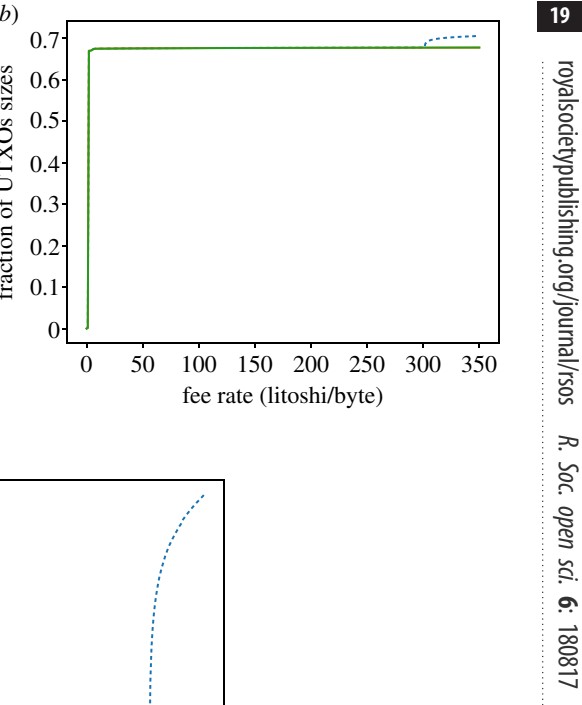

**Figure 14.** Dust and unprofitable analysis for Litecoin. (*a*) Fraction of dust/unprof. UTXOs w.r.t. fee-per-byte rate, (*b*) fraction of occupied space w.r.t. fee-per-byte rate and (*c*) fraction of economic value w.r.t. fee-per-byte rate.

stands out compared to the values from the other two. The explanation of such a huge difference is in the content of that snapshot: the 400K snapshot has an outstanding number of UTXOs holding exactly 1000 satoshis, whereas in the 450 K and 500 K snapshots the UTXOs are more uniformly distributed. In fact, the 54.1% of the unprofitable UTXOs located in the 0–50 range for the 400K snapshot hold exactly 1000 satoshi, whereas for the 450 K and 500 K they represent 18.35% and 14.21%, respectively.

Analysing the general trend of the data for figure 17*a*, we see how the accumulation of unprofitable outputs in the lower fee rates tends to increase over time, which increases the number of outputs that will be hardly spent. Such accumulation is mainly with regard to P2PKH outputs, which as we have already seen in §3 represented around 81% of the outputs in the set.

Similarly, we have analysed the evolution of the UTXO sets of Litecoin by studying 13 snapshots, from block 100 000 up until block 1 300 000 stepping every 100 000 blocks. Unprofitability is very stable, with most snapshots (all but 100K) showing very similar distributions (figure 17*b*).

We have omitted the figures related to Bitcoin Cash since they share the same history with Bitcoin (the snapshots up until 450K are thus exactly the same), and therefore do not provide any new information.

# 6. Related work

As we have seen, the characteristics of the UTXO set can be a key point in cryptocurrencies like Bitcoin, Litecoin and Bitcoin Cash. The size and performance of this set have a direct impact on how the system will perform, and it is thus a focus area where to improve the scalability and efficiency of these cryptocurrencies. For example, transaction generation performance in Bitcoin is greatly influenced by the size of the UTXO set [9].

We can currently find typical statics and simple visualizations of the UTXO set of Bitcoin [10,11], but we are not aware of a more in-depth study and comparison of the UTXO set of significant cryptocurrencies like the one presented in this paper. We believe that knowing the composition and

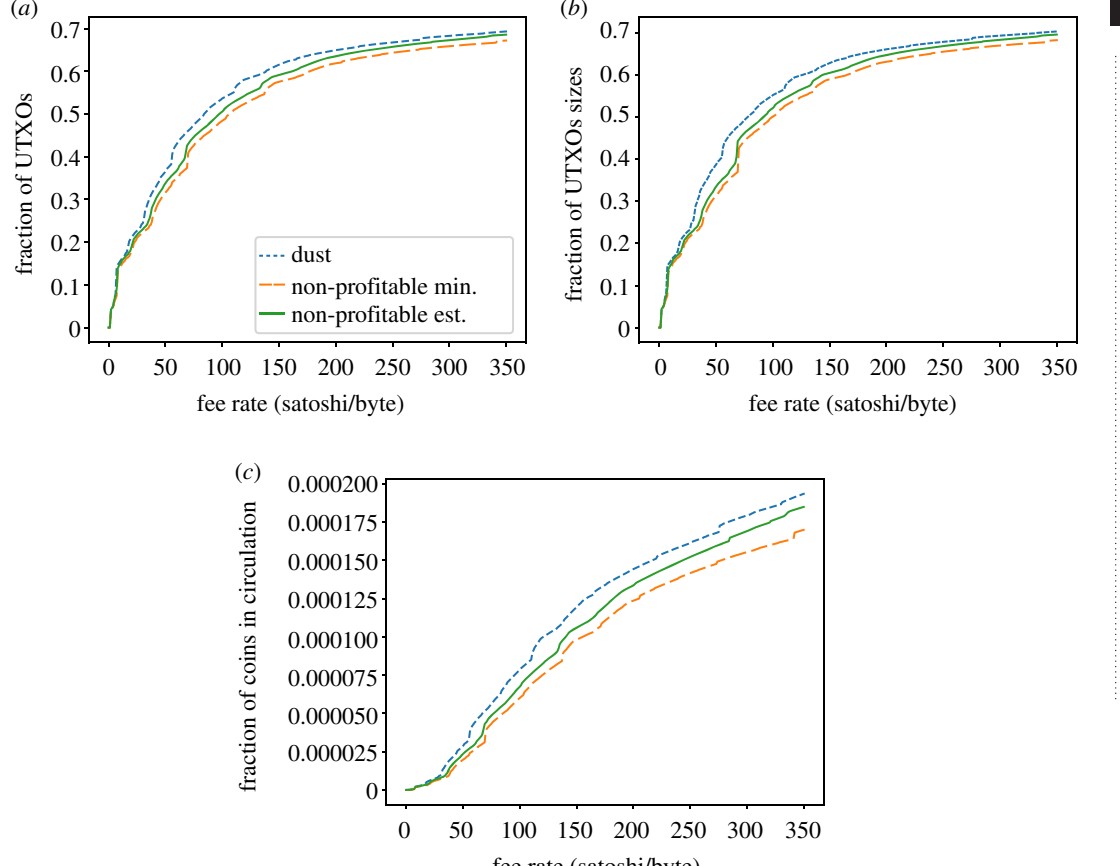

**Figure 15.** Dust and unprofitable analysis for Bitcoin Cash. (*a*) Fraction of dust/unprof. UTXOs w.r.t. fee-per-byte rate, (*b*) fraction of occupied space w.r.t. fee-per-byte rate and (*c*) fraction of economic value w.r.t. fee-per-byte rate.

evolution of the UTXO set will undoubtedly provide the means to better understand it and develop strategies and tools to improve the UTXO set usage, thus enhancing the whole system performance.

The relevance of the UTXO is not new, concerns about its size, composition and performance have been around for some time [12]. These concerns are specially relevant in light of the scalability problems of Bitcoin and are currently an important issue for the future of Bitcoin itself. For instance, Bitcoin Core changed the UTXO set format in version v0.15 in order to improve its performance [13,14]. Both individual users and the whole system will benefit from better management of the UTXO set.

From the user point of view, a strategy of consolidating UTXOs in order to prevent the creation of dust and unprofitable UTXOs in the future (in case of higher fees) has always been considered [15].[15] But such strategies are not easy to generalize. A consolidation will not always reduce the fees for a given user, specially if we cannot anticipate future fee rates. On the other hand, some user will need to maintain a minimum number of UTXOs to be able to generate transactions in parallel. Furthermore, such strategies can undermine the privacy requirements of some users. Given these, sometimes conflicting, constraints and the unpredictability of future fee rates, there is currently no actual strategy for UTXO consolidation.

An important process that directly impacts (and is influenced by) the UTXO set composition and size is the *coin selection* decision performed by wallets [16]. Coin selection is the decision process that a wallet carries in order to choose UTXOs as inputs for a new transaction. Each implementation might use a different coin selection strategy [17]. For instance, if we take a look at Bitcoin, according to [18], several strategies are being used by different wallets. The Bitcoin Core wallet attempts to find a direct match always minimizing the change to be generated. BRD [19] (a popular Android and iOS wallet also known as BreadWallet), uses an FIFO strategy, where the oldest UTXOs from the pool are chosen until the target value is matched. A similar approach is used by Electrum [20] and Mycelium

---

[15]See for instance an example of a consolidation transaction in https://blockchain.info/tx/11ee48127ebf5702 e8883cb829734e444dcc2c4efbdbdfd235726e6acea9fb5c.

**Table 7.** UTXO types in each Bitcoin snapshot.

|  | 100 K | 150 K | 200 K | 250 K | 300 K | 350 K | 400 K | 450 K | 500 K |
|---|---|---|---|---|---|---|---|---|---|
| UTXOs | 71 923 | 1 128 538 | 2 317 071 | 6 802 242 | 10 852 610 | 18 094 009 | 34 818 753 | 44 997 700 | 59 950 252 |
| P2PKH | 19 544 | 1 084 915 | 2 273 595 | 6 731 579 | 10 756 792 | 17 593 948 | 32 158 510 | 36 761 183 | 48 258 713 |
| P2SH | 0 | 0 | 36 | 14 222 | 17 801 | 184 255 | 2 124 567 | 7 658 361 | 11 205 350 |
| P2PK | 52 376 | 43 612 | 42 386 | 38 890 | 41 757 | 41 500 | 41 516 | 50 708 | 69 756 |
| others | 3 | 11 | 1 054 | 17 551 | 36 260 | 274 306 | 494 160 | 527 448 | 416 433 |

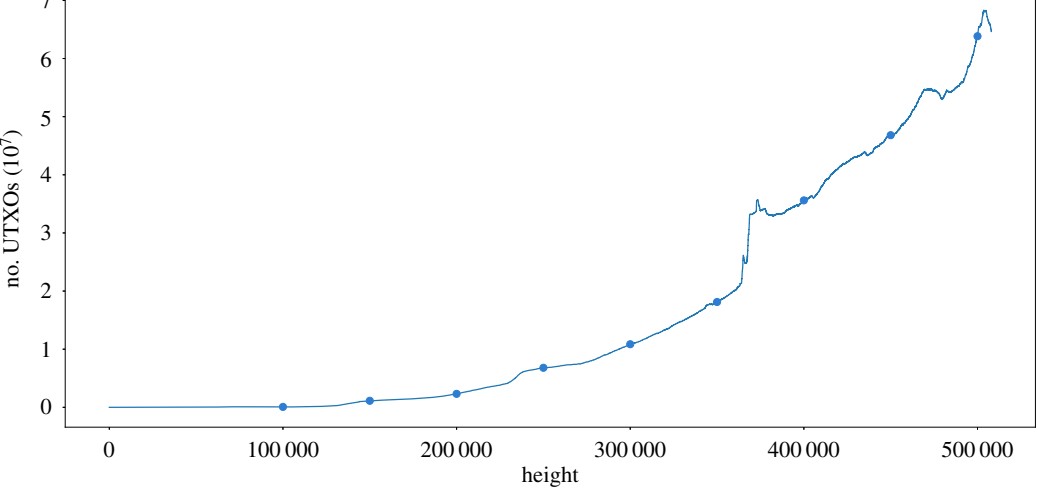

**Figure 16.** Number of UTXOs in each of the snapshots.

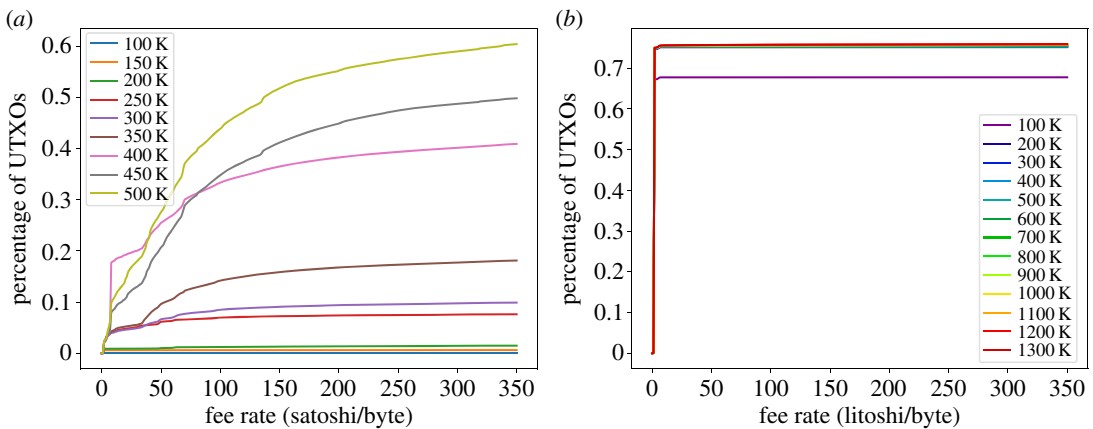

**Figure 17.** Evolution of unprofitable outputs over time. (*a*) Bitcoin and (*b*) Litecoin.

[21] which additionally prunes small-valued UTXOs. The *bitcoinj* library [22] determines a priority metric from the age and value of the UTXOs in order to select new ones. It is by no means clear which strategy is better. Different goals and strategies can be conflicting, such as minimizing the generation of small UTXOs, minimizing the fees for the current and future transactions, or improving the user privacy. Even so, nowadays a common goal shared by all involved parties for the coin selection is actually to prevent the growth of the UTXO set in Bitcoin [18]. In any case, our work introduces new analysis that can help improve these selection strategies.

Following these lines, other proposals such as TXO commitments [23,24] could allow to maintain a smaller functional UTXO set. Similarly, one can think of a two-tier data structure where a UTXO subset containing UTXOs with a low probability of being selected such as dust is kept on disk, while the other UTXOs are kept in memory. We think that the work presented in this paper provides an accurate estimation of such unprofitable UTXOs, which has not been previously considered.

# 7. Conclusion

In this paper, we have presented a detailed analysis of the UTXO sets of the top three UTXO-based cryptocurrencies, that is, Bitcoin, Bitcoin Cash and Litecoin. Our analysis shows that the three cryptocurrencies present both similarities and differences in their UTXO sets.

On the one hand, relevant differences were found in the coin sets of Litecoin with respect to both Bitcoin versions. The two major ones are related to the height where UTXOs are found and the value

they carry. Sixty-seven per cent of Litecoin's UTXOs belong to the first five months of the coin and a similar amount carries just one satoshi. Unlike Litecoin, Bitcoin current UTXOs present more diversity both in their heights and their amounts.

On the other hand, there are some common properties in the UTXO sets of the three cryptocurrencies. First, most transactions with at least one UTXO have indeed just one UTXO. Second, the usage of coins from coinbase transactions is different than the rest of coins. Third, there is a preference for outputs with powers of 10 values. Finally, regardless of the specific metric used to quantify if an output is worth spending (namely, dust or unprofitability), we have found many outputs whose properties (value and size) make them non-economically rational to spend, because spending them results in losing money. These UTXOs represent an important part of the UTXO sets of the three coins (e.g. for a fee of 100 satoshi or litoshi per byte, they represent between 35 and 45% of Bitcoin and Bitcoin Cash UTXOs and 67% of Litecoin's) and occupy also a significant amount of space (similar in percentage). However, they are just a very small amount of the current coin supply (0.01% for Bitcoin and Bitcoin Cash and a negligible percentage of litecoins).

These results indicate the importance of designing proper coin selection strategies in UTXO-based cryptocurrencies. If these strategies are not properly designed and applied, the UTXO sets may end up full of outputs not worth spending. In turn, maintaining these UTXO sets becomes costly, and thus its management may grow into a problem for users with low computational resources available. As a result, incorrect UTXO set management threatens the scalability and the decentralization of UTXO-based cryptocurrencies. Therefore, our work motivates the importance of designing proper coin selection strategies.

This work opens many lines of further research. On the one hand, by describing the composition and properties of the UTXO sets, our analysis can be used to optimize the management of the sets. On the other hand, our work is the first step towards solving the problem of not-worth-spending UTXOs. With this regard, we envision the design of both strategies to disincentivize the creation of dust UTXOs and to incentivize dust consolidation.

Data accessibility. The source code of STATUS can be found in the bitcoin_tools GitHub repository. Data for size estimations is extracted from the Bitcoin, Bitcoin Cash and Litecoin blockchains using BlockSci. The code used for extracting the data can be found in the BlockSci Analysis GitHub repository.

Authors' contributions. C.P.-S. and S.D.-S. developed the STATUS tool used to perform the analysis provided in the paper. All authors discussed which data should be included in the paper and how the results would be graphically presented. All authors proof-read the paper and gave final approval for publication.

Competing interests. We declare we have no competing interests.

Funding. This work is partially supported by the Spanish ministry under grant no. TIN2014-55243-P and the Catalan Agència de Gestió d'Ajuts Universitaris i de Recerca (AGAUR) grant no. 2014SGR-691. C. Pérez-Solà was affiliated with Universitat Autònoma de Barcelona when the first version of the manuscript was done, and with Universitat Rovira i Virgili when the revised version was written. During this later time, she was funded by the Martí-Franquès postdoctoral program from Universitat Rovira i Virgili.

Acknowledgements. The authors thank Joaquín García-Alfaro for the computational resources provided to run some of the experiments of this paper.

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
