## [Reviewer comments · Royal Society Open Science]

Review History

RSOS-180817.R0 (Original submission)

Review form: Reviewer 1

Is the manuscript scientifically sound in its present form?

Yes

Are the interpretations and conclusions justified by the results?

Yes

Is the language acceptable?

Yes

Is it clear how to access all supporting data?

Yes

Do you have any ethical concerns with this paper?

No

Have you any concerns about statistical analyses in this paper?

Yes

Recommendation?

Accept with minor revision (please list in comments)

Comments to the Author(s)

The main contribution of this paper is the analysis of the UTXO sets of Bitcoin, Bitcoin Cash and Litecoin. Methods are introduced to evaluate if an output (of UTXO) is worth spending. In general, the paper is readable and well organized. The paper presents a clear view and structure of UTXO. The analysis indicates that the three cryptocurrencies require relative huge amount of storage complexity in per transaction and height. The experiments in section 4 is quite comprehensive. In Table 3, there are "?". The authors may further explain if those data could be captured dynamically, during the execution of scripts. In figure 10, 11 and 12, there are some "smooth" lines. The authors may further explain the reasons behind.

Decision letter (RSOS-180817.R0)

19-Oct-2018

Dear Professor Herrera-Joancomartí,

The editors assigned to your paper ("Another coin bites the dust: An analysis of dust in UTXO based cryptocurrencies") have now received comments from reviewers. We would like you to revise your paper in accordance with the referee and Associate Editor suggestions which can be found below (not including confidential reports to the Editor). Please note this decision does not guarantee eventual acceptance.

Please submit a copy of your revised paper before 11-Nov-2018. Please note that the revision deadline will expire at 00.00am on this date. If we do not hear from you within this time then it will be assumed that the paper has been withdrawn. In exceptional circumstances, extensions may be possible if agreed with the Editorial Office in advance. We do not allow multiple rounds of revision so we urge you to make every effort to fully address all of the comments at this stage. If deemed necessary by the Editors, your manuscript will be sent back to one or more of the original reviewers for assessment. If the original reviewers are not available, we may invite new reviewers.

- Data accessibility

If you wish to submit your supporting data or code to Dryad (<http://datadryad.org/>), or modify your current submission to dryad, please use the following link:
<http://datadryad.org/submit?journalID=RSOS&manu=RSOS-180817>

- Competing interests

- Authors' contributions

- Acknowledgements

- Funding statement

Please note that Royal Society Open Science charge article processing charges for all new submissions that are accepted for publication. Charges will also apply to papers transferred to Royal Society Open Science from other Royal Society Publishing journals, as well as papers submitted as part of our collaboration with the Royal Society of Chemistry (<http://rsos.royalsocietypublishing.org/chemistry>). If your manuscript is newly submitted and subsequently accepted for publication, you will be asked to pay the article processing charge, unless you request a waiver and this is approved by Royal Society Publishing. You can find out more about the charges at <http://rsos.royalsocietypublishing.org/page/charges>. Should you have any queries, please contact openscience@royalsociety.org.

on behalf of Dr Ghassan Karame (Associate Editor) and Prof. Michael Huth (Subject Editor)
openscience@royalsociety.org

Comments to Author:

Reviewers' Comments to Author:
Reviewer: 1

Comments to the Author(s)

The main contribution of this paper is the analysis of the UTXO sets of Bitcoin, Bitcoin Cash and Litecoin. Methods are introduced to evaluate if an output (of UTXO) is worth spending. In general, the paper is readable and well organized. The paper presents a clear view and structure of UTXO. The analysis indicates that the three cryptocurrencies require relative huge amount of storage complexity in per transaction and height. The experiments in section 4 is quite comprehensive. In Table 3, there are "?". The authors may further explain if those data could be captured dynamically, during the execution of scripts. In figure 10, 11 and 12, there are some "smooth" lines. The authors may further explain the reasons behind.

Author's Response to Decision Letter for (RSOS-180817.R0)

See Appendix A.

Decision letter (RSOS-180817.R1)

26-Nov-2018

Dear Professor Herrera-Joancomartí,

I am pleased to inform you that your manuscript entitled "Another coin bites the dust: An analysis of dust in UTXO based cryptocurrencies" is now accepted for publication in Royal Society Open Science.

As described in our instructions to authors (<https://royalsociety.org/journals/authors/author-guidelines/>), we need the original source files of any figures and tables included with your manuscript, as well as an editable version of your paper for our production processes. Before we are able to proceed further with your manuscript, please ensure that you email me:

A zip folder containing all source files for your manuscript - including a text file of the manuscript (tex, txt, rtf or doc), references, tables (including captions) and figure captions; an electronic file of each figure (EPS or print-quality PDF preferred [either format should be produced directly from original creation package], or original software format); an electronic file of each table.

Please do NOT deposit your manuscript figure or table files in the Dryad repository (<http://datadryad.org/>) -- this may cause unnecessary delays.

Once we have the original files for your manuscript, you can expect to receive a proof of your article in the near future. Please contact the editorial office (openscience_proofs@royalsociety.org and openscience@royalsociety.org) to let us know if you are likely to be away from e-mail contact. Due to rapid publication and an extremely tight schedule, if comments are not received, your paper may experience a delay in publication.

Kind regards,
Andrew Dunn
Senior Publishing Editor
Royal Society Open Science
openscience@royalsociety.org

on behalf of Dr Ghassan Karame (Associate Editor) and Michael Huth (Subject Editor)
openscience@royalsociety.org

Appendix A

November 7, 2018

Dear Editor,

In this document we would like to detail the changes we have performed in the paper to account for the comments we have received.

We have considered each and every one of the comments given by the reviewers in an attempt to make our contribution more insightful. Below we provide detailed responses to the reviewer's comments and explain how and where in the paper we have addressed the comments. We would like to thank the reviewer for his time reading and commenting the paper. The comments have helped us to improve the paper.

Kind regards,

Cristina Pérez-Solà
Sergi Delgado-Segura
Guillermo Navarro-Arribas
Jordi Herrera-Joancomartí

Universitat Autònoma de Barcelona

Reviewer 1

Comment: In Table 3, there are "?". The authors may further explain if those data could be captured dynamically, during the execution of scripts.

Response:

Indeed, maybe the usage of the question mark symbol in the table could cause some confusion to readers. For that reason we have changed such symbol with the notation `var` that probably better indicates the variable nature of such parameter. The main idea that we want to show with this notation is that it is not possible to determine the size of the input that will spend that kind of outputs. However, as the reviewer points out, the moment the output is spent, we can observe the content of its `scriptSig` and, therefore, compute its size. Yet whenever the output is spent, it is no longer an Unspent Transaction Output (UTXO), and it is thus no longer the subject of our analysis. On the other hand, we can indeed use historic data about outputs that have already been spent to estimate the input sizes of such `var` value. This is actually what we do in Section 4.3, where we estimate the unknown sizes given the data we can see in the blockchain, that is, the inputs that have already been revealed.

We have tried to better explain how do we deal with unknowns in input sizes in our study at the beginning of Section 4.c.

Comment: In figure 10, 11 and 12, there are some "smooth" lines. The authors may further explain the reasons behind.

Response:

We have further explained in the paper the smooth lines showed in Figures 10, 11 and 12. We have included the following comments regarding each of the mentioned figures:

- Figure 10: *Before block 158,214, all spent P2PKH outputs have uncompressed public keys in the corresponding inputs, because this was the format being used by bitcoin clients at that time. Consequently, the plot shows a smooth line on the low heights, denoting the constant 64-byte length of public keys being used.*
- Figure 11: *From block $\approx 275,000$ up until block $\approx 480,000$, the average size of inputs is almost constant, because 98.2% of them are 2-out-of-2 and 2-out-of-3 multisig scripts which are 216 and 250-byte long, respectively. However, after this period, P2WSH and P2WPKH scripts encapsulated in P2SH inputs become popular, and because its size is smaller, the average P2SH input script size decreases.*
- Figure 12: *From block $\approx 488,000$ onwards, the plot shows an stable trend. 97% of the redeem scripts are 252, 253 or 254 bytes long, and correspond to 2-of-3 multisignature scripts.*